# Megakaryocytes possess a STING pathway that is transferred to platelets to potentiate activation

Firas El-Mortada[1],*, Karima Landelouci[1],*, Samuel Bertrand-Perron[1], Félix-Antoine Aubé[1], Amélie Poirier[1], Amel Bidias[1] , Georges Jourdi[7,8] , Mélanie Welman[7,8], Michael P Gantier[2,3] , Justin R Hamilton[4,5], Benjamin Kile[6] , Marie Lordkipanidzé[7,8] , Geneviève Pépin[1]

**Platelets display unexpected roles in immune and coagulation responses. Emerging evidence suggests that STING is implicated in hypercoagulation. STING is an adaptor protein downstream of the DNA sensor cyclic GMP-AMP synthase (cGAS) that is activated by cytosolic microbial and self-DNA during infections, and in the context of loss of cellular integrity, to instigate the production of type-I IFN and pro-inflammatory cytokines. To date, whether the cGAS-STING pathway is present in platelets and contributes to platelet functions is not defined. Using a combination of pharmacological and genetic approaches, we demonstrate here that megakaryocytes and platelets possess a functional cGAS-STING pathway. Our results suggest that in megakaryocytes, STING stimulation activates a type-I IFN response, and during thrombopoiesis, cGAS and STING are transferred to proplatelets. Finally, we show that both murine and human platelets contain cGAS and STING proteins, and the cGAS-STING pathway contributes to potentiation of platelet activation and aggregation. Taken together, these observations establish for the first time a novel role of the cGAS-STING DNA sensing axis in the megakaryocyte and platelet lineage.**

## Introduction

Host defence against infection relies on two important systems: the immune and the coagulation systems. Platelets are anucleate cells produced by megakaryocytes and are responsible for blood clotting and haemostasis (Lefrancais et al, 2017). They are also very abundant and play unexpected roles in immune responses (Maouia et al, 2020; Tokarz-Deptula et al, 2021). Hence, platelets are at the crossroads of these critical processes. Platelets possess several immune receptors, including TLRs, that detect microbial molecules and modulate platelet functions, including potentiating their activation and aggregation independently of their canonical transcriptional roles (Ebermeyer et al, 2021). They also interact directly with other immune cells such as lymphocytes, monocytes, and neutrophils. Through these receptors and cellular interactions, they are implicated not only in coagulation, but also in bacterial clearance, defence against viruses and inflammatory responses (Maouia et al, 2020; Tokarz-Deptula et al, 2021).

Interestingly, early literature suggests that platelets respond to double-stranded (ds)DNA, by releasing inflammatory molecules stored in their secretory granules and by aggregating (Fiedel et al, 1979). More recently, it was also observed that platelet activation by DNA leads to increased levels of the cell surface protein P-selectin and the release of the chemokine platelet factor 4 (PF4/CXCL4), two markers of platelet activation and degranulation (Jansen et al, 2017).

The cyclic GMP-AMP synthase (cGAS) is a cytosolic DNA receptor. cGAS detects microbial DNA during infections and nuclear or mitochondrial self-DNA as a result of loss of cellular integrity (Civril et al, 2013; Sun et al, 2013; Motwani et al, 2019; Guey & Ablasser, 2022). Upon DNA sensing, cGAS produces the second messenger, cyclic GMP-AMP (cGAMP), which then binds to the stimulator of IFN genes (STING) adaptor protein (Ablasser et al, 2013; Wu et al, 2013). In cells with a nucleus, activated STING acts as a scaffold protein to recruit TANK-binding kinase 1 and IKKe to allow for the phosphorylation of the transcription factors: IFN-related factor 3 and NF-κB, resulting in the production of type-I IFN (IFN-I) and pro-inflammatory cytokines (Ablasser et al, 2013; Gao et al, 2013a, 2013b; Ahn et al, 2014). Sustained activation of the cGAS-STING pathway underpins the inflammation observed in many pathological conditions, including cardiovascular diseases, ageing, obesity, and neurological disorders (Motwani et al, 2019). Using transcriptomic data from macrophages expressing the S365A STING mutant, specifically deficient for IFN-I signalling, a recent

[1]Groupe de Recherche en Signalisation Cellulaire, Département de Biologie Médicale, Université du Québec à Trois-Rivières, Trois-Rivières, Canada    [2]Centre for Innate Immunity and Infectious Diseases, Hudson Institute of Medical Research, Clayton, Australia    [3]Department of Molecular and Translational Science, Monash University, Clayton, Australia    [4]Australian Centre for Blood Diseases, Monash University, Melbourne, Australia    [5]CSL Innovation, Melbourne, Australia    [6]Monash Biomedicine Discovery Institute and Department of Biochemistry and Molecular Biology, Monash University, Clayton, Australia    [7]Centre de Recherche, Institut de Cardiologie de Montréal, Montréal, Canada    [8]Faculté de Pharmacie, Université de Montréal, Montréal, Canada

Correspondence: Genevieve.pepin3@uqtr.ca
*Firas El-Mortada and Karima Landelouci are shared co-first authors

study has unveiled the IFN-independent STING activation pathways, emphasising the broad implication of this pathway beyond its canonical role (Wu et al, 2020). Among these IFN-I–independent functions of STING, a pivotal role of STING in lethal coagulation during sepsis has recently been reported. Molecularly, this study showed that STING activation in macrophages results in increased cytosolic calcium, which then causes the secretion of tissue factor, ultimately contributing to a pathological coagulation cascade characterised by high levels of D-dimers during sepsis (Zhang et al, 2020). Despite this first evidence of a role of the cGAS-STING pathway in coagulation, whether the cGAS-STING proteins have a role in other cell populations contributing to coagulation, including in platelets, remains to be investigated until very recently. In fact, while revising our article, a new study reported that STING in platelets contributes to thromboinflammation during sepsis, confirming some of the findings we are showing in this study (Yang et al, 2023).

Platelets are released into the vasculature from the fragmentation of cytoplasmic protuberances called proplatelets. Alternate methods of platelet generation, such as from the budding or the rupture of the megakaryocyte plasma membranes, have been observed in isolated studies that still require to be corroborated (Vitrat et al, 1998; Junt et al, 2007; Potts et al, 2020). Whether cGAS is present in platelets and whether the presence of cGAS and STING in platelets relies on their transfer from megakaryocytes remains to be defined.

In this work, we initially investigated the presence and functionality of the cGAS-STING axis in megakaryocytes and platelets. We demonstrated that megakaryocytes and platelets possess the cGAS and STING proteins, which can be functionally activated, albeit differently. As such, although STING drives a tonic response in megakaryocytes, it also potentiates the activation and aggregation of platelets. Taken together, our work demonstrates a previously unrecognised role of the cGAS-STING pathway in megakaryocytes and platelets, and raises important questions as to their functions in thromboinflammation.

## Results

### STING stimulation of megakaryocytes induces a type-I IFN response

Platelets are released from cytoplasmic protuberances of megakaryocytes and have little translation capacity under resting conditions, so we hypothesise that to be present in platelets, the cGAS and STING proteins would first have to be translated in megakaryocytes. In fact, in silico analysis of platelet datasets shows that the mRNAs of cGAS and STING were either not detected, or detected at a very low level in murine and human platelets (Manne et al, 2020; Campbell et al, 2022; Nuhrenberg et al, 2022). STING was, however, detected at the protein level in human platelets (Su et al, 2022), and more recently in mouse platelet lysates (Yang et al, 2023). To investigate the presence of cGAS and STING in megakaryocytes, we relied on in vitro differentiation of cells expressing CD41, an integrin specific to the megakaryocyte's lineage (hereafter named CD41+ cells), isolated from adult mouse bone marrow cells that were

then cultured in the presence of thrombopoietin (TPO) to stimulate their maturation. Consistent with the phenotype associated with mature megakaryocytes, we could observe the formation of pro-platelet structures in a high number of the megakaryocytes (Fig 1A). Mature megakaryocytes were easily identified in our brightfield images by their multilobed nucleus, and the expression of the CD41 integrin at the plasma membrane (Fig 1D–F). When stimulated with DMXAA, a synthetic murine STING agonist (Prantner et al, 2012), we observed the release of IFN-β cytokines (Fig 1B) and CCL5 (RANTES) (Fig 1C).

The unique maturation process of megakaryocytes prompted us to determine the localisation of both cGAS and STING in megakaryocytes. cGAS intracellular localisation varied depending on the cell being analysed. We could detect cGAS in the nucleus, in the cytoplasm, and in bright punctates (Fig 1D). In fact, these punctates were found in the nucleus and in micronucleus-like structures (small nuclear vesicles surrounded by lamin A/C containing or not DNA). These different localisations of cGAS suggest that cGAS localisation is dynamic. The nuclear localisation of cGAS could be consistent with previous studies, showing that during mitosis, cGAS is located at the chromatin where its activity is reduced (Zhong et al, 2020; Helbling-Leclerc et al, 2021). Although megakaryocytes do not undergo mitosis per se, they do replicate their DNA and go through several rounds of abortive mitosis; a process called endomitosis (Vitrat et al, 1998), suggesting that cGAS might be similarly inhibited. Unfortunately, we also discovered that the antibody we used

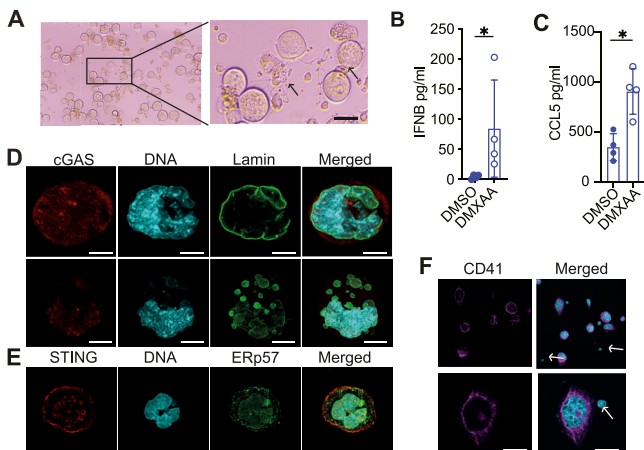

**Figure 1. STING stimulation of megakaryocytes induces a type-I interferon response.**
**(A)** Brightfield images of megakaryocytes isolated using anti-CD41 magnetic beads at day 4 of in vitro differentiation. Enlargement shows the production of proplatelets (black arrows) by mature megakaryocytes. Scale bar: 30 μm. **(B, C)** Cytokine production analysis by ELISA of megakaryocytes stimulated with DMXAA for 5 h (B) and 24 h (C) showing a significant increase in IFNB and CCL5 cytokines, n = 4 independent experiments; each dot represents data from one independent experiment. The graph shows data ± STD, and data were analysed by unpaired two-tailed $t$ tests, *$P$ < 0.05. **(D, E)** Representative immunofluorescence images of megakaryocytes taken by confocal microscopy showing the localisation of cGAS or STING (red), ERp57 or lamin A/C (green), and chromatin (blue), representative of n = 3 independent experiments. Scale bars: 20 μm (D) and 15 μm (E). **(F)** Representative immunofluorescence images of CD41-positive (magenta for CD41+ and blue for DNA) and CD41-negative (only blue) cells taken by confocal microscopy. Representative images of more than n = 3 independent experiments. Scale bars: 30 μm.

stained the nucleus in cGAS-deficient cells, and this non-specific staining changed depending on the batch we received. To our knowledge, this was the only antibody reported in the literature to stain endogenous murine cGAS (Volkman et al, 2019). Fortunately, this background was not present in the cytosolic part of the cells (Fig S1), and a closer look at the staining pattern demonstrated that the nucleus of cGAS-expressing cells has two different kinds of staining: diffused and concentrated in foci, the latest being also present in the cGAS-deficient cells (Figs S2 and S1). These analyses confirm the presence of cGAS in the cytosolic and nuclear part of megakaryocytes, but suggest that the analysis of cGAS in the nucleus will require additional validation through the generation of better antibodies.

We then imaged the ER-resident protein STING, using an antibody we validated (Fig S3), together with the ERp57 protein, a thiol isomerase protein previously reported to be concentrated in the dense tubular system of megakaryocytes and transferred to platelets (Crescente et al, 2016). Interestingly, ERp57 is also known to be located next to the ER membrane Ca(2+) sensor, STIM1, which has been shown to retain STING at the ER (Srikanth et al, 2019). As shown by our image, STING has a similar localisation pattern to ERp57 (Fig 1E). Some areas of colocalisation are also observed, suggesting that both proteins might be transferred to platelets similarly.

## STING stimulation induces ISGs in megakaryocytes

Because even a small number of contaminating cells can impact gene expression analysis experiments, we used the CD41 marker to determine the purity of our cell preparations. As shown in Fig 1F, non-CD41 small cells were contaminating our preparations. To determine whether cGAS or STING stimulation could result in IFN-I response, we conducted a gene expression analysis on CD41-CD42b sorted megakaryocytes (Fig 2A). In line with cGAS activity being reduced in megakaryocytes, DNA stimulation of megakaryocytes increases only to a small extent the expression level of *Ifit1* and *Ccl5*.

However, stimulation of the same cells with DMXAA increased the expression levels of the ISGs measured (*Ifit1*, *Rsad2*, and *Ccl5*) supporting the capacity of STING to induce an IFN-I response in megakaryocytes. After activation, STING translocates from the ER to the Golgi during signalling (Mukai et al, 2016). To complement our study, we stained the DMXAA-treated cells and their control cells with the phospho-(P)-STING antibody, which we previously validated (Fig S4), and the Golgi marker TGN46 antibody (Fig 2B). Image analysis confirms an increase in cells with P-STING foci, with some foci colocalising with TGN46 (Fig 2C). We also noticed that P-STING was present in the control cells, suggesting the basal activation of

**Figure 2. STING stimulation induces the expression of interferon-stimulated genes.**
**(A)** Gene expression analysis of sorted megakaryocytes (based on the expression of CD41 and CD42b) stimulated with DNA + Lipofectamine (DNA) or Lipofectamine only (LF) for 5 h or with DMXAA for 2 h showing the level of specific ISGs (*Ifit1, Rsad2, Ccl5*); each dot represents the data from one experiment. Data are shown as ± STD and were analysed by unpaired two-tailed t tests, $*P ≤ 0.05$ and $**P ≤ 0.01$. **(B)** Representative immunofluorescence images of WT megakaryocytes stimulated or not with DMXAA taken by confocal microscopy showing the localisation of CD41 (yellow), P-STING (red), TGN46 (green), and chromatin (blue). n = 3 independent experiments. Scale bars: 20 μm. **(C)** Quantification of P-STING intracellular foci between WT megakaryocytes stimulated or not with DMXAA. Each dot represents the number of foci in one megakaryocyte. Images were taken from n = 3 independent experiments. Data are shown as ± STD and were analysed by unpaired two-tailed t tests, $**P ≤ 0.01$. **(D)** Representative immunofluorescence image of megakaryocytes taken by confocal microscopy showing the localisation of cGAS (red), lamin A/C (green), and chromatin (blue) in relation to the amount of DNA damage (P-γH2AX) in the four stages of megakaryocyte's maturation (one to four). Scale bars for lanes 1 and 2: 10 μm; and for lanes 3 and 4: 20 μm. **(E)** Quantification of the fluorescence mean intensity associated with the P-γH2AX DNA damage marker; each dot represents the measure for a single megakaryocyte; megakaryocytes were measured from at least n = 2 independent experiments. The graph shows data ± STD, and data were analysed by unpaired two-tailed t tests, $**P ≤ 0.01$ and $****P < 0.0001$. **(F)** Schematic representation of the proportion of megakaryocytes from each stage in relation to cGAS localisation (nuclear, cytosolic, or both).

the pathway; a phenotype that could be driven by the genome instability of megakaryocytes during maturation.

## Dynamic localisation of cGAS during megakaryocyte maturation

To investigate whether megakaryocytes display genome instability during maturation, we imaged megakaryocytes and separated them into groups based on their ploidy and their size (Fig 2D). The first group of cells included the smallest cells (average of 15 μm) and displayed 2 or 3 nuclei; the second group consisted of medium cells (average of 23 μm) and displayed 3–5 nuclei; the third group were larger cells (average of 30 μm) with more than five nucleus; and the last group were very large cells (up to 55 μm) with a multilobed shape. Besides cGAS itself, we chose to image P-γH2AX and the lamin A/C to assess the presence of DNA damage and the integrity of the nuclear membrane, respectively. By doing so, we unveiled a correlation between cGAS localisation and the amount of DNA damage. In fact, in the early stage of maturation, megakaryocytes displayed low levels of DNA damage (P-γH2AX) and no signs of nuclear membrane ruptures, and exhibited a predominantly nuclear distribution for cGAS. However, as megakaryocytes progressed into maturation, they accumulated DNA damage, which coincided with a high proportion of cGAS detected predominately in the cytosol near the plasma membrane and in micronuclei. However, in the last stage of maturation, DNA damage was almost undetected, and the nuclear integrity of the cells was completely lost; thus, it was difficult to assess the localisation of cGAS. Together, these data suggest that genome instability contributes to cGAS activation (Fig 2E and F).

## STING drives tonic activation of ISGs during megakaryocyte maturation

The fact that cGAS was present in micronucleus-like structures, a hallmark of genome instability (Dou et al, 2017; Gluck et al, 2017; Harding et al, 2017; Mackenzie et al, 2017), led us to ask whether cGAS could be basally activated during megakaryocyte maturation. We then posited that if cGAS was basally engaged, we should be able to detect P-STING in WT megakaryocytes, and so IFN-I–stimulated gene expression should be basally higher than in cGAS-deficient cells. As expected, we detected a higher signal of P-STING in WT megakaryocytes than their cGAS$^{-/-}$ counterparts (Fig 3A and C). Furthermore, WT megakaryocytes displayed a higher number of P-STING punctates (Fig 3B), suggesting that the cGAS-STING pathway is basally activated in megakaryocytes. We also treated megakaryocytes with the STING inhibitor C176 during their differentiation (Haag et al, 2018). STING inhibition resulted in a reduction in the expression levels of selected ISGs (Fig 3D). Next, we compared the level of ISGs from WT and cGAS$^{-/-}$ megakaryocytes, isolated from littermate mice. Accordingly, we found that the expression levels of *Ccl5* and *Ifit1* mRNAs were reduced in cGAS$^{-/-}$ megakaryocytes (Fig 3E). Together, these results support the basal activation of the cGAS-STING axis in megakaryocytes that result in the induction of ISGs, as previously reported for other cell population including bone marrow–derived macrophages (Schoggins et al, 2014). Although our analysis suggests that

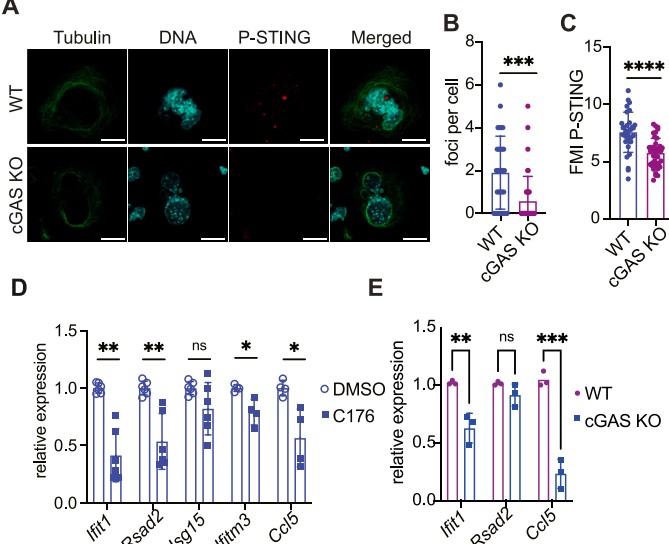

**Figure 3. STING drives a tonic type-I interferon response in megakaryocytes.**
**(A)** Representative immunofluorescence images of WT and cGAS KO megakaryocytes taken by confocal microscopy showing the localisation of P-STING (red), tubulin (green), and chromatin (blue) showing an increase in P-STING fluorescence signal in WT megakaryocytes. **(B, C)** Quantification of P-STING intracellular foci (B) and of the fluorescence intensity (C) for P-STING in WT megakaryocytes compared with cGAS KO; each dot represents a single megakaryocyte image taken from n = 2 biological replicates. The graph shows data ± STD, and data were analysed by unpaired two-tailed *t* tests, ***$P$ ≤ 0.001 and ****$P$ < 0.00001. **(D)** Gene expression analysis of megakaryocytes treated with the STING inhibitor C176 for 48 h (500 nM for 24 h and 1 μM for 24 h) during differentiation showing a significant decrease in the expression levels of ISGs (*Ifit1, Rsad2, Ifitm3,* and *Ccl5*), n = 3 independent experiments in independent experiments. The graph shows data ± STD, and data were analysed by unpaired two-tailed *t* tests, *$P$ < 0.05 and **$P$ ≤ 0.01. **(E)** Gene expression analysis of WT or cGAS KO megakaryocytes after differentiation showing a decrease in the expression levels of selected ISGs (*Ifit1 and Ccl5*). Data shown are from n = 3 independent experiments. Data are shown as ± STD and were analysed by unpaired two-tailed *t* tests, *$P$ < 0.05 and **$P$ ≤ 0.01.

this activation might be triggered by genome instability, we cannot exclude that the activation of STING is driven by in vitro culture of our megakaryocytes.

## cGAS is detected in platelets and produces cGAMP

In a subset of megakaryocytes, cGAS and STING were detected near the plasma membrane. This led us to investigate whether these proteins could be transferred to platelets. To test this idea, we cultured the megakaryocytes until we could detect the formation of proplatelets (usually 4 d). We were able to show that cGAS and STING were present in proplatelets using the B-tubulin and CD41 markers (Fig 4A). These images prompted us to determine whether cGAS and STING could also be present in platelets. To test this, we isolated platelets from WT and cGAS$^{-/-}$ mice, depleted from contaminating leucocytes using CD45-negative selection. Protein analysis showed that both cGAS and STING are present in mouse platelets under physiological conditions, suggesting that inflammation is not necessary for their presence in platelets (Fig 4B). STING was detected in washed platelets by

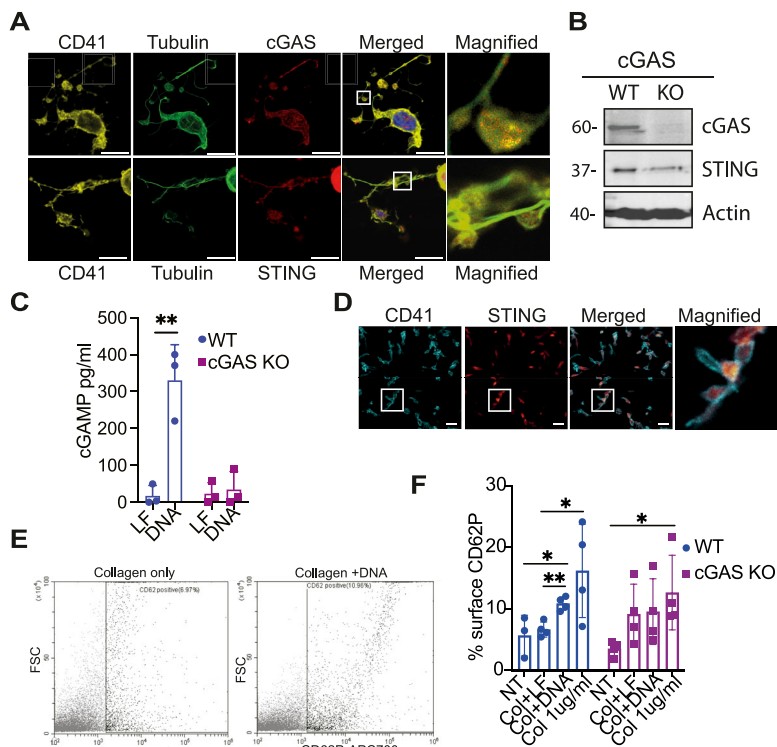

**Figure 4. cGAS and STING stimulation potentiates the activation of platelets.**
**(A)** Representative immunofluorescence image of a megakaryocyte taken by confocal microscopy showing the localisation of CD41 (yellow), cGAS or STING (red), tubulin (green), and chromatin (blue) in proplatelets. Images of megakaryocytes producing proplatelets were obtained from n = 3 independent experiments (average of 5 images per experiment). Scale bars: 25 μm. **(B)** Protein analysis by Western blotting of murine platelets isolated from cGAS WT or KO mice. Representative of n = 3 independent experiments. **(C)** cGAMP production was measured from washed platelets isolated from WT or cGAS KO mice stimulated with DNA + Lipofectamine (DNA) or Lipofectamine only (LF) for 30 min before analysing their lysates by specific cGAMP ELISA. Data shown (data ± STD) are from n = 3 biological replicates; each dot represents a single experiment, and data were analysed by unpaired two-tailed t tests, **P ≤ 0.01. **(D)** Representative immunofluorescence images of mouse platelets taken by confocal microscopy showing the localisation of STING (red) and CD41 (blue). Images of platelets were obtained from n = 2 independent experiments. Scale bars: 5 μm. **(E, F)** Flow cytometry analysis of CD62P surface expression on platelets isolated from WT and cGAS KO mice, stimulated with collagen alone or in combination with DNA. Collagen 1 μg/ml was used as a positive control. Platelets were first gated on FSC, SSC, and CD41 before the analysis of CD62P surface expression. Data are presented as % of platelets with CD62P above the unstimulated control. **(F)** Data shown are from n = 4 independent experiments; each dot represents a single experiment, and data were analysed by unpaired two-tailed t tests, *P ≤ 0.05. **(E)** Representative dot plot of CD62P expression levels for WT platelets.

confocal immunofluorescence, in which we used the integrin CD41 to delineate the plasma membrane (Fig 4D). Unfortunately, the cGAS signal was too weak and could not be detected above the background. However, when stimulated with DNA, platelets isolated from WT mice, but not from cGAS$^{-/-}$ mice, produced a significant amount of cGAMP (Fig 4C), strongly supporting the presence and functionality of cGAS in platelets.

### cGAS activation of washed platelets potentiates their activation

Given that platelets are anucleate, they are not capable of activating the canonical STING-IFN-I signalling in the presence of cGAMP. To investigate whether cGAS and STING stimulation could activate platelets, we relied on the analysis of the platelet activation and degranulation marker P-selectin (CD62P) (Kappelmayer & Nagy, 2017). Using washed platelets isolated from WT and cGAS$^{-/-}$ mice, we measured the level of P-selectin at the surface of platelets stimulated with DNA using flow cytometry. DNA alone could not induce translocation of P-selectin at the cell surface. However, when used in combination with a sub-threshold concentration of collagen, DNA stimulation significantly increased the level of P-selectin at the platelet surface of WT compared with cGAS$^{-/-}$ platelets (Fig 4E and F), suggesting a synergistic effect.

### cGAS and STING stimulation potentiates human platelet aggregation

We investigated whether the activation of platelets by cGAS or STING was conserved between mice and humans and could lead to

platelet aggregation. To test this effect, we isolated platelets from healthy volunteers and confirmed the presence of cGAS and STING by Western blotting using washed platelets (Fig 5A and B). We next assessed the capacity of human platelets to aggregate using different concentrations of collagen, with 1 μg/ml being sufficient to reach maximum aggregation (Fig 5D). In similarity to murine platelets, neither DNA nor cGAMP alone could induce platelet aggregation. However, when using a sub-threshold concentration of collagen in combination with DNA or cGAMP, it led to a rapid aggregation of platelets (Fig 5C and D). Taken together, these results support a new role of the cGAS-STING pathway in the biology of platelets.

## Discussion

This study is, to our knowledge, the first to report that megakaryocytes possess a functional cGAS-STING pathway. This is also the first demonstration that stimulation of the cGAS activates platelets and potentiates their aggregation. Our findings define a novel and non-canonical function for the cGAS-STING pathway.

Using both a pharmacological inhibitor of STING and the genetic depletion of cGAS, we discovered that STING activation using DMXAA results in the activation of ISGs. In contrast, we found that DNA stimulation of megakaryocytes did not stimulate this response. This could be attributed to similar mechanisms to what was reported for the inhibition of cGAS during mitosis (Boyer et al, 2020; Michalski et al, 2020; Zhao et al, 2020; Zhong et al, 2020; Helbling-Leclerc et al, 2021). We observed a difference in the level of P-STING between WT and cGAS$^{-/-}$ megakaryocytes, and measured a reduction of ISGs in

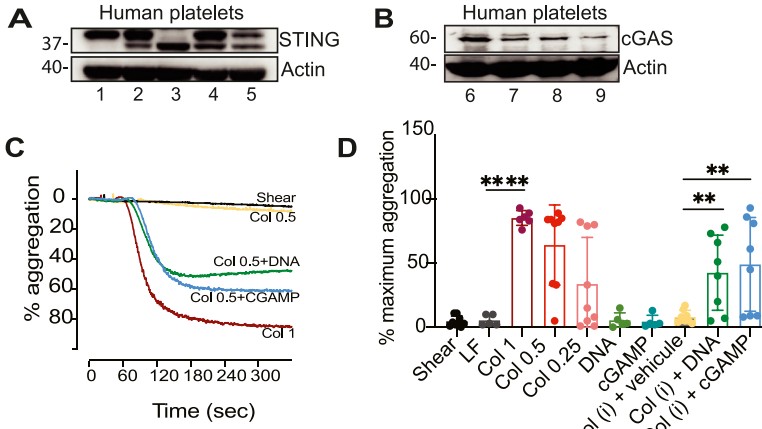

**Figure 5. cGAS and STING stimulation potentiates the aggregation of human platelets.**
**(A, B)** Protein analysis by Western blotting of human platelets taken from five donors (lanes 1–5) for STING and four donors (lanes 6–7) for cGAS. **(C, D)** Human platelet aggregation after treatment with type-I collagen (1, 0.5, or 0.25 μg/ml), DNA, cGAMP, and vehicle alone (Lipofectamine); to test the capacity of DNA or cGAMP to potentiate aggregation, a sub-threshold dose of collagen was determined individually for each patient (col [i]) and used in combination with vehicle, DNA, or cGAMP. **(C)** Representative aggregation curves for one donor (C). Analysis of the maximum aggregation percentage for all donors. **(D)** Data shown are from n= 8 donors (the data from two donors were excluded based on our incapacity to obtain a sub-threshold dose of collagen to perform the potentiation experiment). The graph shows data ± STD, and data were analysed by unpaired two-tailed *t* tests, **$P ≤ 0.01$ and ****$P < 0.0001$.

megakaryocytes treated with the STING inhibitor C176, suggesting that there is basal activation of STING signalling, which was also confirmed using WT and cGAS$^{-/-}$ megakaryocytes. The fact that we detected P-STING punctates in cGAS$^{-/-}$ megakaryocytes, although to a lower extent, suggests that other cGAS-independent mechanisms might be at play. In fact, in cancer cells treated with the DNA damage agent etoposide, nuclear STING activation requires the DNA repair proteins, ataxia telangiectasia mutated and poly-ADP-ribose polymerase 1, and requires IFN-inducible protein 16, but not cGAS (Dunphy et al, 2018). Nonetheless, our results using cGAS$^{-/-}$ megakaryocytes still suggest that cGAS is basally activated in megakaryocytes. Whereas the endogenous causes of the increase in DNA damage remain to be defined, one possible explanation is an increase in mitochondrial ROS, as suggested by a recent study (Poirault-Chassac et al, 2021). Whether these mtROS cause DNA damage and contribute to the translocation of cGAS will need further investigation, but our data support this idea.

Megakaryocytes are also located in the lung vasculature, where they contribute to the production of a pool of blood platelets (Lefrancais et al, 2017). In this study, we selectively characterised megakaryocytes from the bone marrow, but lung megakaryocytes have been suggested to act like antigen-presenting cells (Yeung et al, 2020; Pariser et al, 2021). As such, we could expect cGAS and STING proteins to share a more cytosolic localisation to rapidly alert the cells of invading pathogens (Barnett et al, 2019). In line with that, it was recently shown that SARS-CoV-2 RNA has been detected in lung megakaryocytes (Shen et al, 2021) and that elevated levels of lung megakaryocytes correlate with severe COVID-19 pathology (Bernardes et al, 2020). Given that the cGAS-STING pathway contributes to the antiviral response against RNA viruses, including against SARS-CoV-2 (Sridharan et al, 2013; Sun et al, 2017; Domizio et al, 2022), it would be interesting to determine whether cGAS is involved in the antiviral response of lung megakaryocytes. Accordingly, a study reported that megakaryocyte infections by dengue and influenza viruses do induce a protective type-I IFN (Campbell et al, 2019).

Besides their role in megakaryocytes, our study also reports for the first time that human and mouse platelets possess both cGAS and STING proteins. In our image analysis, we could indeed detect cGAS and STING in proplatelets. How these proteins traffic from megakaryocytes to reach the site of platelet generation is still not defined. STING is a transmembrane protein that is located at the ER and transits through the trans-Golgi, before being degraded into lysosomes or recycled after its activation (Mukai et al, 2016; Lepelley et al, 2020). Based on that, one could easily envision a model in which STING uses the trans-Golgi network and multi-vesicular bodies similar to other membrane-associated receptors such as P-selectin or to the von Willebrand factor found in alpha granules (Chen et al, 2018). While we were revising our article, a new study showed that STING in platelets associates with the SNARE complex to regulate the secretion of alpha granules (Yang et al, 2023).

We have also discovered an unexpected role of cGAS and STING proteins in platelets. Our results show that cGAS and STING stimulation potentiates platelet activation and aggregation. Interestingly, aspirin, an inhibitor of platelet functions, also suppresses cGAS activity (Dai et al, 2019). When activated, platelets release the content of their granules, which include inflammatory molecules, adhesion proteins, serotonin, and small nucleotides (Chen et al, 2018). A similar priming effect to sub-threshold concentrations of classical platelet agonists was reported for TLR9, another DNA receptor (Thon et al, 2012). However, it was later reported that platelets from TLR9 KO mice still aggregate when stimulated with CpG dinucleotides and collagen, suggesting that at least another receptor is involved in DNA sensing (Thon et al, 2012). Our study suggests that this receptor is cGAS, but more experiments are needed to confirm this hypothesis. Together, these studies suggest that DNA stimulation of platelets contributes to both inflammation and clotting.

One of the important questions raised by our study is the biological contexts allowing cGAS and STING stimulation. As discussed above, the recent study reported that one of these contexts is severe sepsis for which STING in platelets contributes to the pathological thromboinflammation (Yang et al, 2023). Further analysis would be necessary to determine whether this activity is cGAS-dependent. In this study, they also show that cGAMP could be detected in the platelets of septic mice. Given that our data suggest that platelets can produce cGAMP, it would be interesting to

determine whether cGAMP is produced by cGAS from platelets or is uptaken from the extracellular environment.

To get activated, platelet cGAS and platelet STING would need to be in direct contact with DNA or with cGAMP. Direct activation of STING could occur during platelet infections of bacteria producing cyclic dinucleotides, or by viruses that engage the cGAS-STING pathway such as *Staphylococcus aureus* or HIV (Youssefian et al, 2002; Corrigan et al, 2011). Platelets are also known to endocytose molecules from their extracellular environment. Whether the endocytosed molecules could reach cGAS or STING would, however, need to be investigated. If possible, inflammatory contexts leading to the release of the cellular content, such as infected cells that rupture (Bertheloot et al, 2021) and circulating free DNA during sepsis (Margraf et al, 2008), could lead to cGAS-STING activation in platelets and contribute to hypercoagulopathy. We also need to emphasise that STING activation in endothelial cells was reported in the context of severe COVID-19, in which high levels of activated platelets are present (Hottz et al, 2020). The outcome of activated platelets interacting with cGAMP-producing endothelial cells remains, however, to be determined. We can also foresee that platelets will be able to directly uptake extracellular cGAMP, as shown for antigen-presenting cells (Ahn et al, 2018; Pépin et al, 2020). Conversely, a scenario in which cGAMP from platelets is transferred to interacting immune cells through protein channels like LRRC8 can also be envisioned (Zhou et al, 2020). In view of the many outcomes resulting from STING activation, we could expect platelet cGAMP to modulate antigen presentation (Wang et al, 2017), autophagy (Saitoh et al, 2009), inflammatory responses (Anastasiou et al, 2021), and even cell death (Gulen et al, 2017) in interacting cells.

In summary, we report for the first time that megakaryocytes and platelets possess a functional cGAS-STING pathway. We have identified a novel role of cGAS and STING in platelets, conserved in humans and mice. Taken together, these observations support a model in which STING activation in platelets potentiates activation and coagulation. This model could contribute to microbial clearance during acute immune responses but may also underpin tissue damage and hypercoagulopathy during uncontrolled immune responses, as seen during sepsis (Yang et al, 2023). Given the growing contribution of megakaryocytes and platelets in immune responses, our findings pave the way to important research avenues.

# Materials and Methods

### Mice

The animal protocol was approved by the institutional review board protocol (*Université du Québec à Trois-Rivieres* [CBSA-ethic certificate 2021-G.P.1]), and guidelines of the Canadian Council on Animal Care were followed. For our studies, we used male and female 8- to 14-wk-old C57BL/6J WT and cGAS$^{-/-}$ mice (B6(C)-Cgastm1d(EUCOMM)Hmgu/J; Jackson Laboratories). Mice were housed in a pathogen-free environment. Mice were sex- and age-matched when directly compared. Blood was collected on mice anaesthetised with isoflurane, and bones were collected after cervical dislocation.

### Megakaryocyte purification and culture

Bone marrow cells were obtained by flushing with DMEM the inside of bones (iliac crest, femurs, and tibiae from both legs of the mice) and centrifuged at 300g for 10 min at 4°C and resuspended in 1 ml of isolation buffer (PBS 1x, sterile, pH 7.2, 0.5% BSA (*BioTech*), and 2 mM EDTA). The cells were centrifuged again at 300*g* for 10 min at 4°C and resuspended in 90 µl of isolation buffer. To isolate CD41$^+$ cells, 10 µl of FcR blocking reagent (*Miltenyi Biotec*) was added to the cell suspension for 10 min at 4°C. Cells were then labelled with 10 µl anti-CD41 antibody (coupled to Biotin or APC—*Miltenyi Biotec*) for 10 min at 4°C, then washed with 2 ml of isolation buffer, and centrifuged at 300*g* for 10 min at 4°C, twice before the addition of 20 µl of Anti-Biotin/APC MicroBeads (*Miltenyi Biotec*) and 70 µl of isolation buffer for 15 min at 4°C. The cells were washed as described before and resuspended in 5 ml of isolation buffer solution. Then, positive selection was carried out using large column cells according to the manufacturer's protocol (*Miltenyi Biotec*). Finally, CD41$^+$ cells were resuspended in growth medium (DMEM with 4.5 g/litre glucose, L-glutamine supplementation, and sodium pyruvate additive accompanied by 10% [vol/vol] heat-inactivated foetal bovine serum and 1% antimycotic-antimycin) supplemented with 50 ng/ml of thrombopoietin and grown for 3–5 d at 37°C in a 5% CO$_2$ and 95% air humidified incubator. When needed, cells were stimulated with 2 µg/ml of a 70-nucleotides synthetic dsDNA, hereafter referred to as DNA (unless mentioned otherwise), or 2 µg/ml of cGAMP complexed with Lipofectamine 2000 (*Invitrogen*) with a ratio of 1:0.5 in FBS-free medium (OptiMEM (1x) with HEPES additive, 2.4 g/litre sodium bicarbonate, and L-glutamine supplementation [*Gibco*]) or with the murine STING agonist DMXAA at 20 µg/ml. For flow cytometry cell sorting, bone marrow cells were obtained as described above. Lineage cells were depleted using Direct Lineage Cell Depletion Kit for mouse (#130-110-470; Miltenyi Biotec) according to the manufacturer's protocol, and the lineage-negative cells were incubated for 48 h in growth medium (DMEM with 4.5 g/litre glucose, L-glutamine supplementation, and sodium pyruvate additive accompanied by 10% [vol/vol] heat-inactivated foetal bovine serum and 1% antimycotic-antimycin) supplemented with 50 ng/ml of thrombopoietin. Before the sort, cells were stained using CD41-VB515 130-122-768 and CD42b-PE 50245268 (clone 1C2; eBioscience) on a Melody cell sorter (BD) using a 100-µm nozzle on a purity mode. Around 10,000 cells per mouse were recovered, and three mice per experiment were used.

### Platelet isolation (mouse)

Blood was collected from the mice from the inferior *vena cava* (IVC) with a 25-gauge needle and a 1-ml syringe containing 100 µl of 3.8% sodium citrate (3.22% citrate, 112.9 mM citrate, 123.6 mM glucose, 224.4 mM sodium, and 114.2 mM hydrogen ions) as anticoagulant, and subsequently transferred to a 1.7-ml microcentrifuge, which is gently inverted several times to ensure proper mixing. The blood solution was then diluted with 1:1 PBS, centrifuged at 130*g* for 15 min at room temperature. All centrifugations were carried out without a brake and with a low acceleration mode. Two thirds of the upper layer consisting of platelet-rich plasma (PRP) was transferred to another 1.7-ml microcentrifuge tube, which was centrifuged at 130*g*

for 5 min to remove leucocyte contamination. This step was repeated, and the supernatant was transferred to a new 1.7-ml microcentrifuge and centrifuged at 1,000$g$ for 10 min with 100 µl of 3.8% sodium citrate to pellet the platelets. Platelets were then resuspended in 90 µl HBSS, pH 6.4, and 5 mM EDTA or Tyrode's buffer, pH 7.35–7.4, depending on the experiment. To deplete residual contaminating leucocytes, 10 µl of biotinylated CD45 anti-mouse antibody (*Miltenyi Biotec*) was added to the platelet solution and incubated for 10 min at room temperature. The platelet solution was then submitted to negative selection by placing the tube on a magnet to bind the CD45$^+$ cells. The supernatant containing the platelets was then transferred to a new 1.7-ml microcentrifuge tube and centrifuged for 1,000$g$ for 10 min to pellet the platelets without leucocyte contamination (platelet purity analysed by FACS was between 97 and 98%). Platelets were then resuspended in HBSS with 5 mM EDTA, pH 6.4, or in Tyrode's buffer for experimentations. When needed, mouse platelets were stimulated with 5 µg/ml DNA and with 0.25 to 1 µg/ml type-I collagen (*Chronolog Inc.*).

### FACS

To measure the surface level of CD62P (P-selectin), washed platelets were stimulated at 37°C under agitation with the indicated agonist for 30 min before adding the collagen for 15 min. Platelets were then stained with the anti-CD41-BV515 and anti-CD62P-R718 antibodies before being fixed for 15 min using 4% PFA, washed, and analysed using a Beckman CytoFLEX cell analyser (four lasers). 20,000 events were analysed per sample per condition. Results were analysed using the company software CytExpert (Beckman).

### Platelet isolation (human)

This study was approved by the Research Ethics Committee of the Montreal Heart Institute (#2017-2154) and was conducted in accordance with the Declaration of Helsinki. Participants were considered healthy if they were aged 18 yr or older, did not require long-term medical therapy, had refrained from drugs known to influence platelet function in the previous 2 wk, had not undergone major surgery in the previous 6 mo, did not have a history of bleeding symptoms, and had platelet counts and haemoglobin levels within the normal reference range. Informed consent was obtained from each participant. Blood was drawn by venipuncture into syringes containing acid citrate dextrose (ACD-A, DIN: 00788139) in a 1:5 volume ratio (ACD/blood). Blood was transferred to 50-ml tubes and centrifuged at 200$g$ for 10 min without a brake, and PRP was collected. Prostaglandin E1 (1 µM) was added to PRP before centrifugation at 1,000$g$ for 10 min. Plasma was removed and discarded. Pelleted platelets were resuspended in Tyrode's buffer (137 mM NaCl, 11.9 mM NaHCO$_3$, 0.4 mM NaH$_2$PO$_4$, 2.7 mM KCl, 1.1 mM MgCl$_2$, and 5.6 mM glucose, pH 7.4). Platelet counts were determined using a Beckman Coulter haematology analyser (D x H 520) and adjusted to a final concentration of 2.5 × 10$^8$/ml. Platelets were allowed to rest at room temperature for 60 min before functional experiments.

### Light transmission aggregometry

Platelet aggregation was measured using a Chronolog aggregometer (Model 700) at 37°C with continuous stirring at 1,200 rpm. Platelet aggregation traces were recorded for 30 min using the AGGRO/LINK8 Software package. Collagen (Chronolog) 0.06–1 µg/ml was used to stimulate platelet. When specified, washed platelets were pre-incubated with DNA (2 µg/ml), cGAMP (2 µg/ml) complexed with Lipofectamine 2000 (*Invitrogen*) with a ratio of 1:0.5 in FBS-free medium (OptiMEM [1x] with HEPES additive, 2.4 g/litre sodium bicarbonate, and L-glutamine supplementation [*Gibco*]), or vehicle for 30 min at room temperature.

### Gene expression analysis

RNA was extracted from cells lysed with RLT lysis buffer using RNeasy Micro Kit (*QIAGEN*) and according to the manufacturer's protocol. RNA was reverse-transcribed using Applied Biosystems High-Capacity cDNA Reverse Transcription Kit. Quantitative (q)RT-PCR was performed using SensiFAST SYBR No-ROX Kit (*Bioline*), and specific primers for genes of interest were used (Table S1). qRT-PCR was performed in an iCycler (*Bio-Rad*) qRT-PCR instrument. Results were normalised on the housekeeping genes TBP and 18S using the ddCT method.

### ELISA

mIFNB was measured from the supernatant of cells stimulated for 5 h with DMXAA using the mouse IFN-beta Quantikine ELISA kit (*R&D Systems*) according to the manufacturer's protocol. mCcl5 was measured from the supernatant of cells stimulated for 24 h with DMXAA using the mouse mCcl5 ELISA kit (*R&D Systems*) according to the manufacturer's protocol. cGAMP ELISA was performed using specific cGAMP ELISA (*Cayman*) on either cell lysates or platelet supernatants after 1 h of stimulation with DNA according to the manufacturer's protocol. For platelet stimulation, C176 was added at a concentration of 1 µM for 30 min before DNA stimulation.

### Microscopy

For live-cell imaging of megakaryocytes, an Optika inverted trinocular LED fluorescence microscope was used. For fixed cell analysis, cells were prepared as follows. Platelets were isolated, fixed with PFA 4% for 10 min, and added on coverslips. For megakaryocyte confocal imaging, CD41$^+$ cells were grown and differentiated directly on coverslips for 1–4 d depending on the experiment. Cells were fixed with 4% PFA. Cells (not platelets) were permeabilised using 0.2% Triton in PBS 1X. Permeabilised cells or platelets were incubated in blocking solution (0.1% Triton with 1% BSA in PBS 1X) for 15 min before their incubation with specific antibodies. Coverslips were mounted on slides with ProLong Gold Antifade mounting reagent (*Thermo Fisher Scientific*). DAPI was used to stain the DNA of megakaryocytes. Images were taken using a Leica SP8 laser scanning fluorescence confocal microscope and analysed with LAS X software from Leica.

## Statistical analyses

Statistical analyses were carried out using Prism 7 (GraphPad Software Inc.). Two-tailed unpaired $t$ tests were used to compare pairs of conditions. Symbols used are as follows: *$P \leq 0.05$, **$P \leq 0.01$, ***$P \leq 0.001$, and ****$P \leq 0.0001$.

# Supplementary Information

# Acknowledgements

We thank Dr Stephane Chappaz and Dr Kate McArthur for help with initial experiments; Melodie B Plourde for training students on the instruments; and Natacha Merindol for help with the FACS. This research was supported by the Université du Québec à Trois-Rivières; the Fonds de Recherche du Québec (FRQ-NT) – Nature et technologies (266767 to G Pépin); the John Evans Leader Canadian Foundation for Innovation (40780 to G Pépin); and the Natural Sciences and Engineering Research Council of Canada (RGPIN-2022-04349 to G Pépin). M Lordkipanidzé is Canada Research Chair in Platelets as biomarkers and vectors (950-232706).

## Author Contributions

F El-Mortada: formal analysis, investigation, methodology, and writing—review and editing.
K Landelouci: formal analysis, investigation, methodology, and writing—review and editing.
S Bertrand-Perron: investigation, methodology, and writing—review and editing.
F-A Aubé: formal analysis, investigation, and writing—review and editing.
A Poirier: investigation.
A Bidias: investigation, methodology, and writing—review and editing.
G Jourdi: investigation, methodology, and writing—review and editing.
M Welman: investigation, methodology, and writing—review and editing.
MP Gantier: resources and writing—review and editing.
JR Hamilton: resources and writing—review and editing.
B Kile: resources and writing—review and editing.
M Lordkipanidzé: resources, supervision, investigation, methodology, and writing—review and editing.
G Pépin: conceptualisation, formal analysis, supervision, funding acquisition, validation, investigation, methodology, project administration, and writing—original draft.

## Conflict of Interest Statement

M Lordkipanidzé has received speaker honoraria from Bayer and JAMP Pharma; has received research grants to the institution from Idorsia; has served on a national advisory board for Servier and JAMP Pharma; and has received in-kind support for investigator-initiated grants from Fujimori Kogyo. There are no other conflicts of interest to disclose.

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
