## [Reviewer comments · Life Science Alliance]

Life Science Alliance

Megakaryocytes possess a STING pathway that is transferred to platelets to potentiate activation

Firas El-Mortada, Karima Landelouci, Samuel Bertrand-Perron, Félix-Antoine Aubé, Amélie Poirier, Amel Bidias, Georges Jourdi, Mélanie Welman, Michael Gantier, Justin Hamilton, Benjamin Kile, Marie Lordkipanidze, and Geneviève Pépin

DOI: <https://doi.org/10.26508/lsa.202302211>

Corresponding author(s): Geneviève Pépin, Université du Québec à Trois-Rivières

Review Timeline:

Submission Date:	2023-06-11
Editorial Decision:	2023-06-12
Revision Received:	2023-11-12
Editorial Decision:	2023-11-13
Revision Received:	2023-11-15
Accepted:	2023-11-16

Transaction Report:

June 12, 2023

Re: Life Science Alliance manuscript #LSA-2023-02211-T

Dr. Geneviève Pépin
Université du Québec à Trois-Rivières
Département de biologie médicale
Quebec
Canada

Dear Dr. Pépin,

Thank you for submitting your manuscript entitled "Megakaryocytes possess a functional STING pathway that is transferred to platelets to potentiate activation" to Life Science Alliance. We invite further consideration of this manuscript at LSA, revised according to your proposed revision plan in response to the latest round of review.

Thank you for this interesting contribution to Life Science Alliance. We are looking forward to receiving your revised manuscript.

Sincerely,

B. MANUSCRIPT ORGANIZATION AND FORMATTING:

Referee #1

This is a revised manuscript by El-Mortada et al.

The study contains intriguing observations, however, important concerns have not been addressed and the study remains preliminary.

That STING is expressed in platelets and supports degranulation is known. An interesting and original aspect in the present study is the examination of cGAS and STING in MKs. They have data showing STING/cGAS in murine bone marrow-derived MKs, and it would have been important to confirm these protein expressions are not driven by in vitro culture. They explain that tissue immunofluorescence could not be performed. Does it mean that they failed to identify STING in vivo, or was it not attempted? It seems important to confirm whether these molecules are indeed transferred to platelets as RNA or protein. Knock-out mice could be used in these experiments to control the specificity of the antibodies in bone marrow, and it would inform on its role in MK differentiation in vivo. Presence of proteins in platelets from these mice at baseline could have been easily verified.

We understand that a recent paper has reported the presence of STING in platelets. However, this paper does not have any data reporting the presence and functionality of cGAS in platelets. On the contrary, they suggest that STING in platelets is activated by cGAMP produced by cells from the microenvironment. We thus believe that our data add important information that can add to or challenge their model. As such our work is complementary to the work of Yang et al. We also believe that data reproducibility is very important in science and thus publishing our data validate some of their findings.

We have performed a protein analysis of blood platelets isolated from cGAS^{+/+} and cGAS^{-/-} mice. We now show that cGAS and STING are present in platelets in physiological conditions in our new figure 4B.

We have not tried to perform tissue immunofluorescence for two main reasons. 1) we do not have the expertise to perform in situ immunofluorescence of bone marrow. This is a technically challenging experiments that not many laboratories mastered. 2) It is not clear whether the expression level of cGAS would be sufficient to be detected by tissue staining. The expression of cGAS in megakaryocytes is low compared to other myeloid cells like monocytes and macrophages and there is a lack of availability of good cGAS antibodies that react specifically with the mouse proteins. In fact, in vitro culture of megakaryocytes is heavily used and accepted in the field given the extremely low number of these cells within the tissues. Importantly, all of our experiments rely on primary cells and not immortalized cell lines.

In line with the previous comment, the authors make this statement in the introduction

"Thus, for cGAS-STING proteins to be present in platelets, they first need to be produced within megakaryocytes prior to being translocated and packaged into platelets." How is it excluded that RNA, not proteins, may be transferred in vivo?

We have analyzed RNAseq data sets and did not find a significant level of neither cGAS nor STING mRNAs which greatly reduces the possibility that these mRNAs are translated in platelets. In addition, we have images showing the presence of both proteins in proplatelets and have western blot showing the presence of cGAS and STING proteins in blood platelets under physiological conditions. Altogether these data support a model in which in physiological conditions, cGAS and STING are transferred to platelets rather than being translated. Whether this model still stand in inflammatory conditions is above the scope of our study.

These sentences were added to the text to reflect our hypothesis:

In fact, in silico analysis of platelets datasets shows that the mRNAs of cGAS and STING were either not detected or detected at very low levels in murine and human platelets (Campbell et al, 2022; Manne et al, 2020; Nuhrenberg et al, 2022). STING was however detected at the protein level in human platelets (Su et al, 2022) and murine platelets (Yang et al, 2023).

There is no demonstration of a role for STING/cGAS in vivo, such as a bleeding phenotype, that would support a role in baseline conditions. There is an interesting literature pointing to a role of IFTM3 and regulation of platelet activation in sepsis -cited by the authors (Campbell et al. JCI 2022). Moreover, STING deficiency in mouse platelets reduces platelet degranulation in sepsis (Yang et al. Immunity 2023). Is the STING/cGAS pathways solely necessary for platelet activation in sepsis/infection, and if so would that be due to its absence in vivo when there are no infectious triggers?

The role of STING in physiological conditions has been well characterized in the paper from Yang et al, Immunity 2023. They found no difference in tail bleeding time, in venous thrombosis, or in platelet turnover. They only saw a slightly prolonged occlusion time in FeCl₃-induced carotid artery thrombosis, strongly suggesting that STING does not contribute to hemostasis in physiological conditions. These bleeding tests have not been performed for the cGAS^{-/-} mice as we do not have the models in place. Setting-up these protocols as well as getting the ethic certificate would take several months. However, we have now confirmed that both cGAS and STING proteins are present in megakaryocytes and in platelets in physiological conditions (new figure 4B). We agree with the reviewer that it would be interesting to test whether inflammatory conditions increase the level of these proteins. Since cGAS is an Interferon-inducible gene (ISG), we would expect cGAS to be increased at least in megakaryocytes. However, we believe that this is out of the scope of this study.

P-STING is present in control MK; could this be driven by the in vitro culture? It is a significant finding made by the authors that cGAS/STING may be activated in basal conditions. Confirmation by in situ staining would support the significance of this

finding.

This is a valid point made the reviewer. As mentioned before, the culture of primary megakaryocytes is the gold standard method in the field and we do not have the method in place to do in situ staining. Furthermore, it is not clear that this would work for P-STING since the signal is already faint in cultured cells. We have added this sentence in the text to acknowledge this possibility.

“Although, our analysis suggest that this activation might be triggered by genome instability, we cannot exclude that the activation of STING is driven by in vitro culture of our megakaryocytes.”

It is unclear how Figure 1D should be interpreted, as the authors explain the MK preparations were contaminated. Is this figure really necessary? This figure could be removed.

We have removed the figure 1D

In platelets, STING is mainly located near the granules (Yang et al. Immunity), while in MKs it is at the ER. Do the authors see STING co-localized with granule components as well in MKs?

To characterize the localization of STING, we have now imaged STING together with PDIA3 (ERp57) which is an ER protein transferred from megakaryocytes to platelets (Crescente et al. Arterioscler Thromb Vasc Biol. 2016). As shown by our new images, STING and PDIA3 display a very similar localization with colocalization area. Both proteins are located near the nucleus and near the plasma membrane. The fact that both proteins are located near the plasma membrane is reminiscent of a transfer to proplatelets.

DNA was used at 2ug/ml while DMXAA was at 20u/ml to stimulate MK. How were the concentrations chosen, are these comparable in terms of molarity?

The concentrations were chosen based on what is used in the literature. They do not have the same molarity (DMXAA 70nM and DNA 45 nM-although we used it at 2ug/ml). However, given that one is a chemical that enter the cells directly and the others need to be complexed with lipofectamine, it is not expected that they are used at the same molarity. We have tested different amount of DNA, but this did not result in increased IFN-I response suggesting that we have reached a plateau.

It is not made clear in the main text that cells were transfected with DNA, which could be an explanation why DNA did not reach TLR9. This should be mentioned. If findings could be confirmed with a more relevant stimulus capable of promoting DNA localization in the cytosol would improve the biological significance.

We have now made it clear in the figures, in the M&M and in the main text that DNA was transfected with lipofectamine. The reviewer is right that using Lipofectamine (LF) can prevent TLR9 signaling, but since our goal was to activate cGAS and not TLR9, we

don't see this as an issue. In addition, it was also reported that platelets isolated from TLR9 KO mice still respond to CpG DNA (Thon *et al.*, 2012) which suggest that findings relating to TLR9 in mice platelets should be analyzed with cautions.

The typos previously identified in Figure 4 remain (collolagen, vehicule)
We are sorry about this and we have corrected these typos.

Referee #2

The authors improved the manuscript considerably, also by better image quality. Not all the experiments that were wishful could be done or were successful. Nevertheless the story is in my eyes conclusive.

Typo in Figure 4: "vehicule", "colollagen"

Explain in the legends the abbreviations eg LF=lipofectamine

Describe in Methods the source of the DNA used for stimulation.

We thank the reviewer and have added the information as requested.

Referee #3

This reviewer appreciates the efforts the authors have made in addressing the queries raised during the first round of review. Unfortunately, however, the manuscript remains premature with little depth in the data. The authors' new model is that MKs have inactive cGAS but active STING and that both cGAS and STING are active in platelets. If appropriately supported by data, this would be an interesting finding. However, the finding that cGAS is inactive in MKs relies on a single type of experiment (Fig 2A) that lacks a positive control and sufficient repeats for statistical analysis. Other concerns are listed below.

We thank the reviewer for raising this point. We have repeated the experiments and confirm that ISGs are induced upon DMXAA treatment. Our results also indicate that ifit1 and ccl5 are slightly induced upon DNA treatment, although to a much lesser extent suggesting a low level of activity for cGAS remains.

We also change the title of the subsection to ***STING stimulation induces an IFN-I response in megakaryocytes***

Major points.

1. Considering the new data in Fig 2A (suggesting that MKs do not respond to DNA), Fig 1D is misleading, because the reader only later finds out that the cells in 1D were not

pure. As such, the authors may want to consider removing Fig 1D. Similarly, the data on basal ISG expression in Fig 3D/E are compromised by the new data in Fig 2A. The signal in 3D/E may well be (at least in part) due to the smaller contaminating cells.

We have now removed the data from figure 1D. However, we think that Figure 3D/E are backed-up by the P-STING imaging studies and as such we think they are worth keeping. It would not be possible to conduct all of the data on sorted megakaryocytes due to the high number of mice needed per experiment. We have however changed the sentence referring to this result in the texts to account for this possibility.

2. The response in Fig 2B/C seems only marginally increased upon DMXAA treatment. STING KO cells should be used in this experiment as a control.

We agree that the response is small. However, as the reviewer can appreciate, the P-STING antibody is very specific (appendix 3). We do not have the STING KO mice to perform this control directly in megakaryocytes.

3. Appendix Figure S1 shows a notable amount of background signal in cGAS-deficient cells. This unfortunately diminishes confidence in the cGAS localization data.

We agree with the reviewer and we have tried to remove this background unsuccessfully. We have looked in the literature to find an alternative antibody and we have tested other methods for the fixation and staining without much improvement. This antibody (cGAS (D3O8O) Rabbit mAb #31659, Cell signaling) has been used by the group of Daniel Stetson to stain endogenous cGAS and they also observed a similar nuclear background. (Volkman HE, Cambier S, Gray EE, Stetson DB. Tight nuclear tethering of cGAS is essential for preventing autoreactivity. *Elife*. 2019). In addition, we have noticed that the background changes depending on the lot of the antibody and the latest lot we received showed a stronger background as demonstrated by new figure appendix 4. Despite these technical challenges, we can have images to show that the detection of cGAS in the cytosol and in the proplatelets is very specific. While there is still an issue with the nuclear background, our data still support the presence and transferred of cGAS from megakaryocytes to platelets.

“Unfortunately, we discovered that the antibody we used also stained the nucleus in cGAS-deficient cells and this background was even higher in the last product we received. Unfortunately, this was to our knowledge the only antibody reported in the literature to stain murine cGAS {Volkman, 2019 #160}. Fortunately, this background was not present in the cytosolic part of the cells (appendix 4), and a closer look at the staining pattern demonstrated that the nucleus of cGAS expressing cells has two kinds of staining, diffuse and bright foci the latest being also present in the cGAS deficient-cells. These analyses confirm the presence of cGAS in the cytosolic and nuclear part of megakaryocytes but suggest analysis of cGAS in the nucleus will require additional validation.”

4. The manuscript contains many statements on novelty and data strength (potent, strong, significant, etc.). Such statements are unnecessary and sometimes unjustified, e.g. in the absence of comparisons.

We have revised the main text to remove these statements when they were not appropriate (in the following sentences the red words were removed)

STING stimulation of megakaryocytes induce a **potent** type-I Interferon response

cGAS is located at the chromatin where its activity is **strongly** reduced

Image analysis confirm a **significant** increase of cells with P-STING foci

strongly suggesting that the cGAS-STING pathway is basally activated in megakaryocytes

STING inhibition resulted in a **significant** reduction

Minor points.

1. Page 3. "cGAS detects microbial DNA during infections and nuclear or mitochondrial self-DNA as a result of loss of cellular integrity (Civril et al, 2013; Pepin et al, 2016; Pepin et al, 2017a; Pepin et al, 2017b; Sun et al, 2013)." For the non-expert reader, please cite one or two recent reviews on cGAS here, instead of self-citing articles that are more than five years old.

We have added two recent reviews and removed the self-citations.

2. Page 4. Please delete "potent" from the first subheading of the results section. This qualifier is inappropriate in the absence of comparative data from other cell types or other stimuli of IFN.

‘‘Potent’’ has been removed

3. Unnecessary statements of significance should be avoided. There are several instances of this, for example on Page 5. Paragraph 1. "...a significant release of IFN- β cytokines (Figure 1B) and Ccl5 (RANTES) (Figure 1C) was measured." Please rephrase: "release of IFN- β (Figure 1B) and CCL5 (RANTES) (Figure 1C) was observed."

As we mentioned in point 4, we have removed the ‘‘statements of significance’’

4. Fig 1D/E. mRNAs need to be shown in italics.

These figures were removed

5. Fig 1G. The RTN4 staining adds very little information as it was not done simultaneously with the STING staining.

We have replaced these images by a new representative image showing PDIA3 (ERp57) in co-staining with STING.

6. Fig 1H. Please explain in the legend what the arrows point to.

This has been done

7. Fig 4E. Collagen (not Colollagen)

This has been changed

8. Some English language editing is required.

We have had the manuscript review by a native speaker.

November 13, 2023

RE: Life Science Alliance Manuscript #LSA-2023-02211-TR

Dr. Geneviève Pépin
Université du Québec à Trois-Rivières
Département de biologie médicale
3351 Bd des Forges
Trois-Rivières, Québec G8Z 4M3
Canada

Dear Dr. Pépin,

Thank you for submitting your revised manuscript entitled "Megakaryocytes possess a STING pathway that is transferred to platelets to potentiate activation". We would be happy to publish your paper in Life Science Alliance pending final revisions necessary to meet our formatting guidelines.

- Supplemental Figures should be referred to as such, rather than as "Appendix Figure S..."
- please add callouts in the text for the Supplementary Figures
- please list the Supplementary Figure legends right after the main figure legends in the main text file
- Appendix Figure S5 should be relabeled as Table S1
- please add scale bars to Figures 1A and 2D
- please add sizes next to the blots in Figure 4B

A. FINAL FILES:

B. MANUSCRIPT ORGANIZATION AND FORMATTING:

Sincerely,

November 16, 2023

RE: Life Science Alliance Manuscript #LSA-2023-02211-TRR

Dr. Geneviève Pépin
Université du Québec à Trois-Rivières
Département de biologie médicale
3351 Bd des Forges
Trois-Rivières, Québec G8Z 4M3
Canada

Dear Dr. Pépin,

Thank you for submitting your Research Article entitled "Megakaryocytes possess a STING pathway that is transferred to platelets to potentiate activation". It is a pleasure to let you know that your manuscript is now accepted for publication in Life Science Alliance. Congratulations on this interesting work.

DISTRIBUTION OF MATERIALS:

Again, congratulations on a very nice paper. I hope you found the review process to be constructive and are pleased with how the manuscript was handled editorially. We look forward to future exciting submissions from your lab.

Sincerely,
